# Exploring the Therapeutic Potential of Elastase Inhibition in Age-Related Macular Degeneration in Mouse and Human

**DOI:** 10.3390/cells12091308

**Published:** 2023-05-03

**Authors:** Soumya Navneet, Carlene Brandon, Kit Simpson, Bärbel Rohrer

**Affiliations:** 1Department of Ophthalmology, Medical University of South Carolina, Charleston, SC 29425, USA; 2Department of Healthcare Leadership and Management, Medical University of South Carolina, Charleston, SC 29425, USA; 3Department of Neurosciences, Medical University of South Carolina, Charleston, SC 29425, USA; 4Ralph H. Johnson VA Medical Center, Division of Research, Charleston, SC 29425, USA

**Keywords:** elastin, alpha 1 anti-trypsin, age-related macular degeneration, mouse models, MarketScan^®^ Commercial Claims and Encounters Database

## Abstract

Abnormal turnover of the extracellular matrix (ECM) protein elastin has been linked to AMD pathology. Elastin is a critical component of Bruch’s membrane (BrM), an ECM layer that separates the retinal pigment epithelium (RPE) from the underlying choriocapillaris. Reduced integrity of BrM’s elastin layer corresponds to areas of choroidal neovascularization (CNV) in wet AMD. Serum levels of elastin-derived peptides and anti-elastin antibodies are significantly elevated in AMD patients along with the prevalence of polymorphisms of genes regulating elastin turnover. Despite these results indicating significant associations between abnormal elastin turnover and AMD, very little is known about its exact role in AMD pathogenesis. Here we report on results that suggest that elastase enzymes could play a direct role in the pathogenesis of AMD. We found significantly increased elastase activity in the retinas and RPE cells of AMD mouse models, and AMD patient-iPSC-derived RPE cells. A1AT, a protease inhibitor that inactivates elastase, reduced CNV lesion sizes in mouse models. A1AT completely inhibited elastase-induced VEGFA expression and secretion, and restored RPE monolayer integrity in ARPE-19 monolayers. A1AT also mitigated RPE thickening, an early AMD phenotype, in HTRA1 overexpressing mice, HTRA1 being a serine protease with elastase activity. Finally, in an exploratory study, examining archival records from large patient data sets, we identified an association between A1AT use, age and AMD risk. Our results suggest that repurposing A1AT may have therapeutic potential in modifying the progression to AMD.

## 1. Introduction

Age-related macular degeneration or AMD is a leading cause of blindness among the elderly and is characterized by progressive central vision loss due to photoreceptor degeneration in the macula [1,2]. AMD can be of two types, wet and dry. Dry AMD, the most prevalent form of AMD, involves RPE degeneration followed by photoreceptor cell death [3,4,5]. Wet AMD is the less prominent form of AMD but is the leading cause of AMD-related blindness [5]. Pathological neovascularization into the retina, RPE cell death, and photoreceptor degeneration are the major characteristics of wet AMD [4,5]. Currently, much are still uncertainties about the mechanisms involved in AMD pathogenesis, and there is no cure for AMD. 

Abnormal turnover of elastin, a critical extracellular matrix (ECM) protein in the eye, has been reported in the pathogenesis of AMD [6,7,8,9]. Elastin is a major component of Bruch’s membrane (BrM), an ECM layer situated between the RPE and choriocapillaris in the eye. Thinning/fragmentation of the BrM’s elastin layer (EL) has been reported in AMD [7], and its reduced integrity corresponds to sites of CNV lesions in wet AMD. Gene polymorphisms in elastin [10,11,12,13], and High Temperature Requirement A Serine Peptidase 1 (HTRA1), a protein with elastase-like properties [14,15] are prevalent among AMD patients, and the most significant associations are reported in wet AMD cases. HTRA1 is multifaceted protease which play critical role in ECM remodeling including the elastin turnover [16,17,18]. Overexpression of HTRA1 has been shown to induce polypoidal choroidal vasculopathy and wet AMD like phenotypes in mouse models ([19,20]. Likewise, elevated levels of serum elastin-derived peptides (EDPs) [21] and elastin autoantibodies [22] have also been reported in wet AMD patients. However, the exact role of abnormal elastin turnover in AMD is not yet clear. 

While a major focus for elastin in AMD is the EL of BrM, elastin has been reported in the cornea, conjunctiva, choroid, sclera, muscle tendons and meninges [23], as well as retina [24] and lamina cribrosa [25]. Elastin turnover is mediated by enzymes with elastase-like activity and their respective inhibitors. We have published a review article on “elastin turnover in ocular diseases: A special focus on age-related macular degeneration”, that highlights the multitude of enzymes other than neutrophil elastase (elastase 2), cathepsin L and HTRA1 with elastase-like activity present in ocular tissues [8].

Here we explore the potential effects of general elastase activity in models of AMD and examine a few hypotheses for pathology beyond elastin degradation itself. The generation of serum EDPs, which is expected to lead to the generation of elastin autoantibodies, might mediate antibody-dependent cell-mediated cytotoxicity or complement dependent cytotoxicity and tissue damage. Alternatively, neutrophil elastase is known to cleave protease-activated receptor 2 (PAR2), leading to its activation and down-stream signaling [26], including vascular endothelial growth factor (VEGF) secretion [27]. We report that significantly increased retinal and RPE elastase enzyme activity is associated with AMD-like pathology in mouse and cell-based models. Although elastin degradation was identified, the generation of anti-elastin antibodies was animal model dependent. However, in all models, VEGF was found to be upregulated consistently, and elastase exposure increased VEGF secretion from polarized ARPE-19 monolayers involving PAR2 signaling. Overall, these data suggest that in addition to altering the ECM in the posterior pole, elastase may also increase VEGF production and secretion from the RPE, thereby decreasing RPE barrier function and increasing angiogenesis. Based on these observations we hypothesized that inhibition of elastase activity might be protective in AMD. We used alpha-1 antitrypsin (A1AT), a serine protease inhibitor, which inactivates elastase, in mouse models of AMD. A1AT reduced both the number and size of choroidal neovascularization (CNV) lesions. In a mouse model in which HTRA1, a serine protease with elastase activity [20], is overexpressed specifically in the RPE, A1AT prevented the thickening of RPE/BrM. Finally, an exploratory analysis of archival human data was performed that identified an association between A1AT use and AMD risk among subjects with emphysema.

## 2. Results

### 2.1. AMD Pathology Increases Retinal Elastase Activity in Mouse Models of Wet AMD

As abnormal elastin turnover is reported in patients with wet AMD [8], we investigated whether elastase enzyme activity is altered in mouse models of wet AMD. Pathological or age-related upregulation of elastase enzyme activity can lead to mature elastin fiber fragmentation in tissues generating soluble elastin fragments [17,20,28]. Please note, that as there are many enzymes with elastase activity present in mouse and human RPE [8], and it is unclear which particular enzyme, or which class of enzymes contribute(s) to the abnormal elastin turnover reported, we focus here on general elastase activity. JR5558 mice (JR for short) are a spontaneous wet AMD mouse model which develops choroidal neovascularization (CNV) type III in a VEGFA-dependent manner [29,30]. Here we confirmed that JR mice develop pigmentary changes in the fundus (Figure 1A) and CNV lesions can be identified in fluorescein angiogram images (Figure 2A). Compared to the wild-type (WT) control mice, JR mice had reduced retinal thickness, mainly due to the thinning of outer nuclear layer (ONL) and RPE layer (Figure 1B). Notably, we found increased elastase activity in the retinas of JR compared to the age-matched WT control mice (Figure 1C). Consistently, the retinas of another CNV mouse model, in which CNV is induced by laser photocoagulation of BrM (causing CNV type I), also had significantly elevated elastase activity in the retina compared to the controls (Figure 1D). No significant change in elastase activity was found in the RPE/choroid fractions in either of the two mouse models.

### 2.2. Cultured RPE Cells Derived from Human-AMD Patients or Mouse Eyes with Wet AMD Pathology Exhibit Elevated Elastase Activity

RPE cells are one of the sources of elastases in the eye [8]. We investigated elastase secretion in primary mouse RPE cells isolated from JR and WT mice (Figure 1E–G) and also in iPSC derived-RPE cells from AMD patients and control healthy subjects (Figure 1H,I). Stable mouse and human RPE monolayers were established on transwell plates to allow for polarized secretion, placed into serum-free media, and cell culture supernatants were collected from both apical and basal compartments after 48 h. The apical media from JR RPE monolayers contained significantly increased elastase activity compared to the WT controls (Figure 1F), and there was no significant difference in elastase activity basally. As oxidative stress is a driver in AMD pathology, we treated WT mouse RPE monolayers with 200 μM H_2_O_2_. Oxidative stress significantly increased apical elastase activity in treated WT RPE cells compared to untreated controls (Figure 1E). When analyzing intracellular elastase activity, significantly more enzyme was retained inside the RPE cells isolated from JR compared to the WT RPE cells, and H_2_O_2_ treatment increased intracellular elastase activity only in JR RPE (Figure 1G), but not in WT RPE cells. The results collectively suggest that AMD-like pathology alters the phenotype of RPE cells, leading to increased intracellular and extracellular elastase activities in these cells. Elevated secretion of elastase directed specifically towards the apical side of the RPE correlates with increased retinal elastase activity observed in the intact mouse models (Figure 1C,D). To provide relevance to human AMD, iPSC-RPE cells developed from fibroblasts of dry and wet AMD patients secreted significantly more active elastase towards the apical side compared to the non-AMD controls (Figure 1H). However, no significant difference in elastase activity was found in the basal supernatant (Figure 1I); and intracellular elastase activity was not assessed.

### 2.3. Altered Elastin Turnover and Antibody Levels in Mouse Models of CNV

In wet AMD patients, serum level elastin fragments and autoantibodies are increased [21,22]; and antibody-dependent cell-mediated cytotoxicity or complement-dependent cytotoxicity and tissue damage, both of which require antibodies, have been proposed to play a role in pathology [6,8]. We have shown mouse CNV is augmented by complement activation and autoantibody binding [31]. Here we quantified the amount of elastin fragmentation (i.e., tropoelastin levels) in JR mouse retinas and RPE/choroid to investigate whether the increased elastase activity in those tissues affects elastin turnover. Markedly, and consistent with the results showing increased elastase activity in the retinas of JR mice, the JR mice had increased retinal tropoelastin levels (Appendix A), but no significant difference was found in the RPE/choroid tissues (Appendix A). Increased soluble retinal tropoelastin levels might lead to increased levels of anti-elastin antibodies, similar to that reported in AMD [9,22]. In laser-CNV mice, increased serum levels of IgG anti-elastin antibodies could be detected one month after laser treatment when compared to the age-matched controls (Appendix A); no significant difference was found in the IgM antibody levels (Appendix A). Surprisingly, we found significantly reduced IgG and IgM serum anti-elastin antibody levels in JR mice compared to the age-matched controls (WT [C57BL/6J] and JC [Crb1^rd8^ homozygous mutant]) (Appendix A), and no detectable amounts of antibodies were found in the RPE/choroid or retinal tissues of JR mice (Appendix A). However, these findings were due to an overall reduction in total Ig antibody levels in JR mice compared to the age-matched controls (Appendix A), indicating that JR mice are immune-suppressed. These results suggest that JR mice may not be the right model to study antibody-mediated complement-dependent cytotoxicity (CDC) or antibody dependent cell-mediated cytotoxicity (ADCC) in wet AMD. However, the comparison of both mechanisms should be further explored in laser-induced CNV in C57BL/6J mice.

### 2.4. A1AT, an Elastase Inhibitor, Ameliorates Pathology in Wet AMD Mouse Models 

To further explore the role of elastase in wet AMD pathogenesis, the therapeutic role of A1AT, an elastase inhibitor (Appendix A) was investigated in JR and laser-CNV mice. A1AT was administered intraperitoneally, using weekly administration (100 mg/kg). 

In JR mice, CNV lesions start to develop by 3 weeks of age; with CNV lesion number and vessel leakage peaking at P30, the lesion area by P90 [29,32], and retinal degeneration and ONL thinning starting as early as P30 [29]. Here we showed that JR mice exhibit RPE depigmentation in fundus images and OCT analysis revealed ONL, RPE, and total retinal thinning by 4 months of age, indicative of ECM degradation (Figure 1A,B). When JR mice were treated with A1AT starting at P21, followed by OCT imaging and fluorescein angiography (FA) by P35, FA indicated significantly reduced CNV lesions and leakage in A1AT-injected mice compared to controls (Figure 2A,B). Consistent with this observation, the OCT results showed that ONL, RPE, and total retinal thinning were prevented in A1AT-injected JR mice (Figure 2C–E).

The same beneficial effect of A1AT was found in the laser-induced CNV mouse model as well. CNV was induced at 8 weeks of age, and animals were dosed weekly starting a day before the laser treatment. Volume intensity projections in OCT images were used to quantify the CNV lesion areas as previously published [33] on days 7- and 21 post-CNV induction. CNV-sizes were significantly reduced in A1AT-injected mice compared to the PBS-injected controls at both time points (Figure 2G,H). Our results from these two angiogenesis mouse models demonstrate that A1AT reduces the size of lesions in both type I and III CNV.

### 2.5. Increased Elastase Activity Is Associated with Increased VEGF Levels and Decreased RPE Monolayer Integrity

Neutrophil elastase-mediated injury has been shown to lead to the generation of diffusible VEGF fragments [34], and increased VEGF release is associated with pathological angiogenesis and CNV [35,36]. Overall, VEGF protein, secreted from the RPE or released from the extracellular matrix can bind to receptors present on both the RPE or the endothelial cells, leading to RPE barrier function loss and endothelial cell migration and neovascularization, suggesting that elastase activity might indirectly contribute to wet AMD-like pathology. Stable ARPE-19 monolayers switched to serum-free media, were exposed to neutrophil elastase (NE) on their apical surface and VEGFA mRNA and protein levels were measured after 24 h. Elastase exposure significantly increased VEGF mRNA levels in RPE cells (Figure 3B) and upregulated VEGF protein secretion (Figure 3A). Elastase exposure increased the levels of VEGF in a dose-dependent manner in both the apical and basal supernatants, suggestive of an elastase-VEGF axis that could initiate or contribute to RPE barrier breakdown and CNV in wet AMD. Consistently, VEGFA protein levels were significantly elevated in JR mouse retina extracts compared to the age-matched controls (Figure 3C,D). Under pathological conditions, RPE cells (Figure 1) and infiltrating immune cells such as macrophages, neutrophils, and microglia can all increase elastase levels in the retina (reviewed in [8]), and may contribute individually or together to the increased VEGF levels and wet AMD pathogenesis. While our data implicates the RPE in VEGF production, in vivo, other sources such as Muller glial cells [37] or microglia [38] should not be excluded.

NE-induced VEGF secretion from RPE cells was further investigated mechanistically. As we have shown in the past [39], VEGF secretion involves a VEGF-receptor-mediated feedback loop. Here, 1 h pretreatment with a VEGF receptor inhibitor nintedanib (ND) partially reduced the NE-mediated VEGF release from RPE monolayers (Figure 3F). NE has been shown to activate the protease-activated receptor-2 (PAR2) by cleaving the self-tethered ligand [26], and PAR2 activation has been shown to mediate VEGF production [40]. Blocking PAR2 with the blocking peptide ENMD-1068 prior to NE application partially reduced NE-induced VEGF release from RPE cells (Figure 3G). Finally, NE cleaves complement C3 [41], and complement activation increases VEGF secretion from RPE cells [39,42]. C3a and C5a are the major bioactive complement components that are known to promote CNV in a VEGF-dependent manner [43]. However, ELISA measurements demonstrated that NE treatment in RPE resulted in significantly reduced C3a and C5a levels compared to the non-treated controls (Figure 3H,I), an effect that could be reversed by A1AT. C3a peptide cleavage by NE has been reported previously [44] and might in part explain those results. Overall, our results suggest that elastase-induced VEGF release from the RPE is independent of C3a or C5a receptor stimulation, but it is partially mediated via the activation of PAR2 and VEGF receptors. 

As shown above, A1AT completely inhibited the elastase-induced VEGF secretion from RPE cells, both basally and apically (Figure 3A), and in JR mice, A1AT significantly reduced VEGF levels in the retina (Figure 3C,D). Apical VEGF stimulation has been shown to result in loss in RPE monolayer integrity [45]. Here we show that NE exposure significantly reduced ARPE-19 monolayer integrity and increased RPE layer permeability as assessed by FITC-dextran permeability, which was inhibited by A1AT (Figure 3E). These results suggest that A1AT can reverse or block the RPE permeability phenotype in AMD. 

### 2.6. A1AT Reduces RPE-BrM and Outer Retinal Thickening in HTRA1 Overexpressing Transgenic Mice

HTRA1 is a serine protease with elastase activity [20] and its gene polymorphism is a major risk factor for AMD [14,46,47]. The transgenic mouse models overexpressing HTRA1 protein exhibit AMD-like pathology including ECM changes [17,20,48], and aged-related type III neovascularization [48,49]. Specifically, in a mouse model with RPE specific human HTRA1 overexpression, BrM EL damage, RPE atrophy, photoreceptor degeneration, and polypoidal choroidal vasculopathy (PCV) are reported by 11 months of age using electron microscopy [20]. In the same mouse model, using OCT, we found significant outer retinal changes at 6 months of age (Figure 4). These HTRA1 mice have significantly increased elastase activity in the RPE/choroid tissue (Figure 4C) but not the retina (Figure 4B). Consistently, increased elastase activity was detected in primary RPE monolayer cultures derived from these mice, both intracellularly, as well as in the apical and basal supernatants (Figure 4A). Increased levels of soluble tropoelastin protein were also detected in the RPE/choroid tissues of HTRA1 mice (Figure 4D) consistent with increased elastase activity in the RPE. 

OCT analysis using automated segmentation, revealed significant thickening of the outer retinal layer, which encompasses inner segments, outer segments and RPE (Figure 4F) in HTRA1 mice at the age of 6 months compared to age-matched CD1 control mice. The main contributor to this change is the RPE-BrM complex (Figure 4E). No significant changes were found in ONL thickness (Figure 4G) or overall fundus appearance (Appendix A) at this age. Administration of A1AT, which was confirmed to be able to inhibit the elastase activity of the human HTRA1 protein (Appendix A), starting at 4 months mitigated outer retinal and RPE thickening in HTRA1 transgenic mice (Figure 4E,F) when analyzed at 6 months, suggesting that A1AT can prevent the early outer retinal changes triggered by increased HTRA1 activity. 

At 11 months of age, HTRA1 mice have elevated levels of VEGF [20]. Here we show that using an ex-vivo sprouting assay according to a previous protocol [50], that RPE/choroidal tissues derived from 6–8 month old HTRA1 mice generated larger endothelial sprouts compared to the CD1 controls (Appendix A). The VEGF receptor inhibitor nintedanib completely inhibited sprouting in both HTRA1 and control tissues. Importantly, incubation with A1AT reduced the increased sprouting in HTRA1 but not control tissues. These results add to the body of data presented in this study that increased elastase activity participates in VEGF-mediated damage at the posterior pole.

### 2.7. Preliminary Study on Emphysema Patients, Examining an Association between A1AT Use and AMD Risk

We explored the association between A1AT exposure and AMD in humans using large archival billing data, and hypothesized that A1AT therapy delays the onset of AMD. Our rationale was that in humans, the specific therapy for the treatment of A1-related lung disease and emphysema, expected to affect 40–60,000 people in the US, is replacement therapy. A1AT is administered by weekly intravenous infusion and, until other drugs become available, is considered ongoing and lifelong. The design was a retrospective cohort study using the MarketScan® Commercial Claims and Encounters Database, a database that contains complete records for all dispensed prescription drugs, outpatient visit diagnoses, and hospital admission for each patient, and that is organized by insurance coverage (privately insured age <64 years and Medicare patients age >65 years). We extracted all prescription records for A1AT, and comparison cohorts of patients were constructed from patients without any use of A1AT who also had a diagnosis of emphysema, at a ratio of 1:5. Post-match standardized differences are provided in Appendix A (Appendix A) as are the demographic composition of the matched analysis cohorts (Appendix A).

While the Medicare group was on average 13 years older than the commercially insured patients and had a slightly greater proportion of male patients, patients in both cohorts had a mean of about two years of exposure to A1AT. We used multivariable logistic models controlling for age and time in the study and were not able to differentiate between the two types of AMD in the models due to the small number of wet AMD patients in the cohorts (Figure 5). Finally, the commercially insured group was divided into two subgroups based on exposure length (lower 75% and top 25%).

The multivariable model estimates show a reduced risk of AMD for two of three patient groups defined by age and exposure to A1AT. In the Medicare cohort, we uncovered a 15% reduction in AMD risk (OR: 0.846), in the largest subgroup of the commercially insured patients (75%) a 29.8% reduction (OR: 0.702), while those (25%) with very long exposure to A1AT show a 31.3% (OR: 1.31) increased risk of AMD compared to controls (Figure 5). It is clear that both patient age (row 3) and the length of time in the study (row 4), as well as exposure over many years are modifying effects. Importantly, the model correctly classified patients as having AMD or not based on risk in 76–83% of the time (Model C statistic), increasing the confidence in the data. Overall, the data is indicative of A1AT being protective in the cohort that is younger and requires less treatment, whereas with increasing age and treatment need, A1AT deficiency leads to increased risk of AMD.

## 3. Discussion

A plethora of studies have indicated significant associations between AMD and abnormal elastin turnover [6,8,9,21,22]. However, it’s not clear yet whether the abnormal elastin turnover is a driving force in AMD pathogenesis or is just a bystander effect. In this study, we aimed to determine whether elevated levels of elastase, the enzymes degrading elastin, might have direct roles in AMD-like pathogenesis. In two mouse models of wet AMD (JR5558 and laser-induced CNV) and a model of dry AMD (HTRA1 transgenic mouse), elastase activity was found to be significantly increased in a tissue-specific manner, and in all three models, inhibition of elastase activity by A1AT was found to reduce pathology. While ECM remodeling might be a driver of disease, two additional mechanisms were explored. First, the generation EDPs and the presence of elastin autoantibodies, which might mediate antibody-dependent damage, could not be consistently demonstrated, but is worth exploring further in the individual models. Second, in all three models, upregulated elastase enzyme activity was correlated with an increase in VEGF production, and in RPE monolayers, neutrophil elastase exposure led to increased VEGF secretion partially via the protease activated receptor 2 (PAR2). Relevance to human disease was provided by demonstrating increased levels of elastase activity in supernatants of RPE cells differentiated from iPSC cells from both wet and dry AMD donors. In addition, an exploratory study suggests an association between AMD risk and A1AT exposure in emphysema patients.

### 3.1. Elastase Activation and Its Downstream Effects

Alpha-1 antitrypsin is a protease inhibitor produced primarily by hepatocytes in the liver and inhibits neutrophil elastase activity and other proteases such as HTRA1. As such, A1AT’s main role is to protect tissues and matrices from proteolytic damage. Some of these effects have been discussed above in the context of elastase activity. In addition, anti-angiogenic and anti-inflammatory effects of A1AT have been reported [51,52,53,54]. Its protective role as an anti-inflammatory agent has been reported in two retinal degenerative models [55,56]; A1AT ameliorated retinal degeneration in the rd1 and diabetic mice. In the rd1 mice, a model of retinitis pigmentosa and photoreceptor degeneration, gene and protein levels of A1AT are reduced. A1AT augmentation therapy was found to mitigate retinal pathology and photoreceptor degeneration by reducing inflammation and microglial activation [56]. In diabetic mice, A1AT reduced Tnf-α levels and ameliorated retinal thinning and ganglion cell loss [55]. A1AT also suppressed Tnf-α and MMP12 production in cigarette smoke-exposed macrophages and inhibited thrombin and plasmin activities in cell-free systems [51]. Therapeutic effects of A1AT can be due to its direct hydrophobic interactions with other molecules or its ability to bind to specific receptors or could be due to its protease inhibitory effects [57]. It has been shown that A1AT can reduce LPS-induced Tnf-α and IL8 production from human neutrophils independent of its elastase inhibitory effect [53]. 

Here we added to this body of work by showing that A1AT can result in protection in AMD-like pathology. Our results indicating a significant increase in VEGF secretion from RPE cells exposed to neutrophil elastase confirming the pro-angiogenic potential of elastase in the retina. A pro-angiogenic role for elastase has already been reported in retinal pathology. Increased NE can cause endothelial cell permeability in diabetic retinopathy, a retinal degenerative disease that involves pathological angiogenesis [58]. Additionally, macrophage elastase can contribute to subretinal fibrosis, an end-stage complication of wet AMD [59], and increased protease activity of MMP2/9 has been associated with upregulated VEGF release from human RPE cells [35]. Our results indicate that upon elastase exposure, RPE cells significantly increase VEGF secretion towards both the apical and the basal side. Basal VEGF may increase sprouting of the choriocapillaris contributing to CNV (Appendix A). Co-incubation of RPE cells with NE and A1AT completely inhibited the NE-induced VEGF release from ARPE-19 monolayers (Figure 3A) and inhibited monolayer integrity loss (Figure 3E). The cell-based studies correlated with those obtained in animals. Previous studies [29,32,60] together with our results (Figure 1A,B) indicate that JR mice, a spontaneous wet AMD mouse model, develop late-stage wet AMD phenotypes including type III CNV, and ONL thinning within 1–3 months of age, in a VEGF dependent manner. Administration of A1AT starting at 3 weeks of age significantly reduced the number and size of CNV lesions and ameliorated the outer nuclear layer thinning in these mice. Moreover, A1AT administration significantly reduced type I CNV lesions in the laser-induced CNV mouse model, confirming its therapeutic potential in preventing pathological angiogenesis in all forms of wet AMD. 

Finally, protective effects of A1AT were explored in a transgenic mouse overexpressing human HTRA1 in the RPE specifically. By 11 months of age, the HTRA1 transgenic mouse exhibits degradation of the BrM EL and thickening of the outer collagenous layer of BrM, followed by RPE atrophy, photoreceptor death, and type III CNV [20]. We found a significant thickening of the outer retina and RPE-BrM complex in the same mouse model already by 6 months, prior to developing the more severe phenotypes such as ONL thinning and CNV. Outer retinal and RPE thickening has been identified as an early marker of AMD that precedes severe visual deficits and CNV in patients [61,62]. A1AT injection ameliorated the outer retinal and RPE-BrM thickening in HTRA1 transgenic mice. The potential mechanism leading to tissue thickening, involving deposition of collagen IV [63], has been discussed above. 

### 3.2. Elastase Activity and A1AT in Human Cells and Patients

We have analyzed iPSC-RPE cells derived from fibroblasts of dry, wet and control donors. Interestingly, we found increased levels of elastase activity in the apical supernatants of RPE cells from both wet and dry AMD donors when compared to controls, while the levels were unchanged on the basal side. In unpublished data (Moreira and Rohrer), we find that when grown as monolayers, RPE cells from subjects without AMD appear to exhibit higher TER levels than those derived from subjects with AMD and that their TER levels were negatively correlated with the ratio of VEGF/PEDF. These results are consistent with the observation in ARPE-19 cell monolayers that NE exposure increases VEGF secretion.

Based on these results as well as the animal studies, we performed an exploratory study, testing the hypothesis that there is an association between A1AT exposure and AMD in an exclusive group that all suffer from emphysema, using the MarketScan® Commercial Claims and Encounters Database, analyzing those that are privately insured (age <65) and those covered under Medicare (age >65). While the odds ratios were not statistically significant but only trended, the direction of change based on exposure age, and years in the study suggest some interesting conclusions. With A1AT exposure the AMD risk reduction is 15% in the Medicare group, and 29.8% in the younger, privately insured cohort. In contrast, in those subjects in the top 25% of exposure level, which can be considered the sickest patients, AMD risk is increased by 31%. When analyzing the contribution of age, in the Medicare cohort, for each additional year, the risk of AMD increases by only 9% per year. In contrast, in the younger cohort, for each additional year of age, the risk of AMD increases by 79% per year, or for each additional year of treatment, the risk of AMD increases by 98% year. This might explain why in the top 25% of exposure level subjects, the risk for AMD caught up with them, increasing the risk for AMD despite A1AT exposure; and why the protective effect of A1AT in the older population overall (Medicare) is less than that in the younger group. Our model and observations are novel and important. While the three groups are different and show different AMD risks, the model captures between 76 and 83 % of AMD patients. The next steps will be to examine the characteristics of the A1AT exposure measure in larger samples of patients to examine the statistical significance and to explore the effecting modifiers and correlations with aging and other control variables. In addition, the genotypes for A1AT (SERPINA1) should be taken into consideration. 

It is of interest to note, that a microarray gene expression and linkage analysis study performed to identify novel genes and pathways associated with AMD, identified SERPINA1 as a gene associated with the RAR-related orphan receptor alpha (RORA) and HTRA1 network [64]. In addition, a manuscript currently on medRxiv, analyzing Million Veteran Program together with five other cohorts, identified associations with PiZ (rs28929474) and PiS (rs17580) alleles of SERPINA1 with AMD. Surprisingly, PiZ was found to be protective for AMD, while PiS increased its risk, requiring further investigation. However, while increased HTRA1 protein has been identified in iPSC-RPE cells from donors with homozygous risk at the 10q26 locus with AMD pathogenesis [18], as well as in aqueous humor of nAMD patients [65], and inhibition of HTRA1 with an anti-HTRA1 antibody has been suggested as a therapeutic avenue in geographic atrophy [66], a recent study using donor tissues showed that mRNA levels for HTRA1 are reduced in the RPE, but not in the retina or choroid of donors with homozygous risk at the 10q26 locus. Thus, further work on HTRA1 is required and the interaction with SERPINA1 (A1AT) should be evaluated.

### 3.3. Conclusions

In conclusion, our results suggest that A1AT can reverse the outer retinal alterations that occur during early AMD and can also ameliorate the severe wet AMD retinal phenotypes by inhibiting the pathological VEGF increase and CNV progression. The anti-angiogenic role of A1AT in the wet AMD retina could be due to its elastase inhibitory effect. Incidental exposure to A1AT in human patients shows protection in two of three patient groups against AMD development. Considering the resistance to existing anti-VEGF drugs among many wet AMD patients [67], repurposing A1AT could be a novel, multi-pronged therapeutic strategy for AMD. 

## 4. Materials and Methods

### 4.1. Animals and In Vivo Procedures 

Animals. C57BL/6J and JR5558 mice were purchased (Jackson Laboratory, Bar Harbor, ME) and colonies established at MUSC. HTRA1 overexpressing transgenic mice were generously provided by Dr. Fu (Baylor College of Medicine) [20], corresponding controls (CD1) were purchased (Charles River Laboratories, USA) and colonies established. Animals were housed under a 12:12 h, light: dark cycle with access to food and water ad libitum, and all experiments were approved by the Medical University of South Carolina Institutional Animal Care and Use Committee and performed following the Association for Research in Vision and Ophthalmology statement for the use of animals in ophthalmic and vision research.

CNV lesions. Choroidal neovascularization was induced in 8 weeks old C57BL/6J mice. Mice were anesthetized (with xylazine and ketamine, 20 and 80 mg/kg, respectively), the pupils were dilated (2.5% phenylephrine HCl and 1% atropine sulfate) and the laser lesions were induced using an image-guided laser photocoagulation system (Micron 3, Phoenix research labs) (300 mW, 70 ms) to generate 4 laser spots around the optic nerve of each eye. Bubble formation at the site of the laser burn was used as the inclusion criteria for successful Bruch’s membrane rupture. 

Optical Coherence Tomography (OCT). OCT was performed using SD-OCT, Bioptigen® Spectral Domain Ophthalmic Imaging System (Bioptigen Inc., Durham NC). For OCT, mice were anesthetized with ketamine/xylazine before imaging, the pupils were dilated (2.5% phenylephrine HCl and 1% atropine sulfate) and eyes were kept moist using GenTeal (Alcon) lubricant throughout the procedure. CNV lesion sizes were measured based on individual pixel sizes (μm × μm) from OCT volume intensity projections as described previously [33,68]; retinal layer thicknesses were measured in cross sections using the auto segmentation with InVivoVue™ Diver software (Bioptigen).

Fundus Imaging and Fluorescein Angiography. Fundus images and fluorescein angiography were obtained using the Phoenix Micron III retinal imaging microscope. After anesthesia, pupil dilation, and eye lubrication as described above, mice were administrated subcutaneously 10–20 μl of fluorescein dye (AK-FLUOR 10%, Akorn Inc, IL, USA) followed by rapid acquisition of images over a ~4 min time frame. Care was taken to place the optic nerve in the center for imaging. Fluorescence intensities of the entire fundus were quantified using ImageJ (NIH) without background subtraction [69].

A1AT treatment. A1AT (Zemaira, CSL Behring) was obtained through the MUSC University pharmacy, solubilized in PBS, and injected weekly at the efficacious dose of 100 mg/kg intraperitoneally according to previous publications [55,56].

### 4.2. RPE Cultures

Primary mouse RPE cultures. 9–10 days old pups were sacrificed humanely. RPE tissues were collected, and the resulting cells were plated in a cell culture flask in Alpha MEM medium (Fisher 12571-063) supplemented with N1 supplement (Sigma N6530), GlutaMax (Fisher 35050-061), non-essential amino acids (Sigma M71145), taurine (Sigma T0625), triiodothyronine, hydrocortisone (Sigma), penicillin/streptomycin and 10% FBS. Cells were cultured for a week in the flask before lifting and plating into 12-well cell culture inserts. The cells were maintained at 37 °C and in 5% CO_2_ for 3–4 weeks. After reaching a stable monolayer, cells were transferred to phenol-free and serum-free media, Alpha MEM (Gibco, 41061-029). Transepithelial resistance (TER) was checked to assure the monolayers had sufficient resistance before experiments (around 200 Ω × cm^2^), using an EVOM volt-ohmmeter (World Precision Instruments). Some cultures were treated 3× every 48 h with 200 µM H_2_O_2_ (Sigma) in serum- and phenol-free media. Supernatants were collected from the apical and basal chambers at the end of the treatments and then placed immediately at −80 °C. To collect the cells, 100–200 µL accutase (Gibco) was added per well at room temperature (RT) for 5–10 mins. Cells were spun down, the medium aspirated, and cells snap-frozen followed by storage at −80 °C until use. 

Human iPSC-derived RPE cultures. Skin samples from dry and wet AMD patients and age-matched unaffected controls (ages 60–89) were collected under the approval of the MUSC Institutional Review Boards (IRB 2000020671) and work was performed in adherence to the tenets of the Declaration of Helsinki. Fibroblasts were used to generate iPSC cells [70] followed by differentiation into RPE-iPSC using a published protocol [71]. Please note that the characterization of the cells, including pigmentation, monolayer formation and transepithelial resistance measurements, was published elsewhere [70,72]. Cells were plated at 2.5 × 10^5^/mL onto vitronectin-coated 6-well transwell inserts in E8 medium (Gibco, A1517001) containing Rock inhibitor Y-27632 (Tocris 1254) and 5% FBS. For the detection of elastase activity cells were maintained in serum and phenol-free medium (MEM alpha medium supplemented with B27, glutamax, MEM non-essential amino acids, taurine, and penicillin/streptomycin (Gibco, Thermo Fisher Scientific)). Monolayer status was confirmed by TER (around 200 Ω × cm^2^) and supernatants collected from the apical and basal compartments every 48 h and stored at −80 °C. Supernatants were concentrated (Amicon centrifugal filters with molecular weight cutoff [MWCO] 3kDa) before the experiments.

ARPE19 cell cultures. ARPE-19 cells (ATCC® CRL-2302™; American Type Culture Collection, Manassas VA) from passages 20–30 were expanded in DMEM (Gibco/ThermoFisher Scientific) containing D-Glucose (4.5 g/L), L glutamine, sodium pyruvate (110 mg/L), 1× Anti-Anti (Gibco) and 10% FBS. Confluent cells were trypsinized and equal cells were seeded on 24 mm transwell inserts (Costar, 0.4 µm pore size). After reaching confluency the FBS percentage in the media was gradually reduced to 2%, and then to 1% to promote cell differentiation and tight junction formation as described by us earlier [33,45]. Polarity of the monolayers has been documented by our collaborator [45] as well as by us [42,73] in part based on polarized secretion of growth factors and complement components. Integrity of the RPE monolayer after 1 month of culture was assessed by TER, and wells with TER of ~40–45 Ω × cm^2^ were used for the experiments. For assessment of elastase activity or treatment with chemicals and reagents, cells were exposed to serum-free media.

Chemicals, reagents and procedures used in vitro. A1AT (Zemaira, CSL Behring LLC, IL, USA) was used at a concentration of 1 mg/ml. Neutrophil elastase/human leukocyte elastase (CK828; Elastin Products Company, Missouri) was used at a concentration of 2.5–5 μg/ml. Nintedanib (Tocris Bioscience, Bristol, UK), an anti-angiogenic drug targeting the receptor tyrosine kinases VEGFR, was used at 1 μM. To induce oxidative stress in primary RPE treatments, 200 μM H_2_O_2_ from Sigma was used. The PAR2 antagonist ENMD-1068 (Sigma-Aldrich) was used at 100 μg/ml concentration. Recombinant human HTRA1 (2916-SE-020) was purchased from R&D systems.

Treatment regimens. For all three RPE cell types, monolayers were placed in their respective serum-free media prior to experimentation. Elastase activity was assessed in supernatants 48 h after media exchange. For experiments with VEGF receptor and PAR2 inhibitors cells were pretreated with inhibitors for 1 h before treating with NE. NE and inhibitor treatments were applied apically. Treatments with NE and the respective inhibitors was performed for 24 hours prior to collecting the supernatants for respective readouts. 

FITC dextran trans-epithelial permeability assay. Trans-epithelial permeability of ARPE-19 monolayers were measured using Fluorescein isothiocyanate–dextran (FD20S, Sigma) as posted on *Protocol Exchange* (https://doi.org/10.21203/rs.2.20495/v1 accessed on 28 April 2023). FITC dextran was resuspended in phenol free DMEM and 500 μl of the FITC dextran solution (1 mg/ml) was added to the apical compartments of each well, and 1.6 ml fresh phenol free media to the basolateral compartments. Plates were incubated in the dark at room temperature for 20 min. The basolateral media were collected, vortexed and 100 μl aliquots added into individual wells of a black 96 well plate. 100 μl phenol free DMEM was added into the blank wells. Fluorescence readings were taken at 490 nm/520 nm excitation/emission using a Bio Tek Synergy plate reader.

### 4.3. Elastase Activity Assay

Elastase activity was analyzed according to the manufacturer’s protocol using the EnzChek™ Elastase Assay Kit (E12056, Invitrogen), an assay that is well-cited in the literature [20,74,75]. To measure elastase activity from apical and basal cell culture supernatants, media were collected, and filter-centrifuged to concentrate before the assay. Tissue and cell lysates were prepared in 1× reaction buffer containing 0.01% Triton X. After brief sonication the lysates were centrifuged at 16,000× *g* for 10 min. Supernatants were collected and immediately assayed to detect elastase activity, and all assay reactions were normalized with equal protein. Pierce BCA protein assay kit (Thermo Scientific) was used for protein quantifications. Elastase activity was detected based on the fluorescence readings obtained at 485ex/535em from a Biotek Synergy H1 microplate reader. 

### 4.4. VEGF ELISA

VEGF levels were determined using the VEGF_165_ Quantikine ELISA Kit (DVE00, R&D systems) according to the manufacturer’s protocol. The cell supernatants collected were centrifuged at 500× *g* for 5 min to remove any cell residues, and 200 μl per well was used for the analysis. Absorbance values were taken at 450 nm with a wavelength correction at 540 nm. VEGF protein levels were calculated in picograms after curve fitting using the standards provided in the kit.

### 4.5. C3a and C5a ELISA

C3a and C5a peptides were detected using ELISA kits purchased from Abcam (ab279352) and Invitrogen (BMS2088), respectively. Apical cell culture supernatants were analyzed according to the manufacturer’s protocol. Briefly, for C3a detection, 100 μl of standards and samples were added into the wells and incubated for 2.5 h at room temperature. For C5a ELISA, 200 μl cell culture supernatants were added to the sample wells. After incubation and washing wells were incubated with biotinylated antibodies for 1 h followed by washing and incubation with streptavidin for 1 h, followed by incubation with TMB substrate for 40 min, and finally, the STOP solution was added, and the absorbance readings were taken immediately at 450 nm. The amounts of C3a and C5a proteins were quantified after curve fitting using the peptide standards provided in the kit.

### 4.6. Western Blotting

Tissues and cells were lysed in a buffer containing 50 mM Tris HCl (pH 7.4), 5 mM EDTA, 150 mM NaCl, 0.5% NP-40, 0.5% sodium deoxycholate, 1% SDS and 1× protease-phosphatase inhibitor cocktail (Thermo Fisher Scientific) [76]. 10–20 μg of protein were subjected to SDS polyacrylamide gel electrophoresis at 90–120 V. Gels were transferred to nitrocellulose membranes at 90 V for 1 h. Membranes were blocked in 5% milk in 1× TBST (1 h at RT) and were incubated in primary antibodies overnight in 5% milk in TBST buffer at 4 °C. After washing 3× in TBST, membranes were incubated with HRP-conjugated anti-rabbit or anti-mouse IgG antibody for 1–2 h at RT. The protein bands were visualized using ECL Western blot detection system (Thermo Fisher Scientific, Rockford, IL, USA). Membranes were reprobed with anti-GAPDH antibody, which served as the loading control. Primary antibodies used and dilutions: Anti elastin (PR-387, Elastin Products Company, Owensville, Missouri, USA 1:200); Anti VEGFA (A5708, Abclonal, Woburn, MA, USA, 1:500); Anti GAPDH (2118S, Cell Signaling, Danvers, MA, USA, 1:1000). Secondary antibodies used: HRP linked anti-Rabbit IgG (NA9340V, Sigma Millipore, ST. Louis, MO, USA, 1:2000); HRP-linked Anti-mouse IgG (7076, Cell Signaling, 1:2000). For detecting the total IgG antibodies in retinal and RPE/choroid tissues the membranes were incubated with HRP conjugated goat anti-mouse IgG secondary antibody (A4416, Sigma) overnight at 4 °C. Bio-Rad image lab was used for protein band quantifications.

### 4.7. Real-time PCR

Quantitative RT PCR was performed on cells collected in TRI Reagent (Millipore Sigma, ST. Louis, MO, USA) for RNA preparation. 2 μg RNA was converted into cDNA using the iScript™ Reverse Transcription Supermix (Bio-Rad, Hercules, CA, USA) according to the manufacturer’s protocol. qPCR was performed using SsoAdvanced™ Universal SYBR® Green Supermix (Bio-Rad). PCR conditions were set at 95 °C for 10–15 s and 56–60 °C for 30 s; melt curve analysis confirmed the purity of the end products. Relative VEGF mRNA levels were normalized to that of GAPDH. Ct values (ΔΔCt) were used to quantify the gene expression levels [77]. Analyses were done in triplicates. PCR primers were obtained from Integrated DNA Technologies (Coralville, IA). VEGFA: forward: 5’GCAGCGACAAGGCAGACTAT3’, reverse: 5’AACCAACCTCCTCAAACCGT3’; GAPDH: forward 5’GATGCTGCCCTTACCCCG3’, reverse: 5’ATCCGTTCACACCGACCTTC3’.

### 4.8. Human Data Analysis 

The design was a retrospective cohort study using the MarketScan® Commercial Claims and Encounters Database (calendar years 2010–2020) to determine the odds of having AMD (wet or dry) in the context of A1AT use. This database contains complete records for all dispensed prescription drugs, outpatient visit diagnoses and hospital admission for each patient. The data are provided in different file groups organized by insurance coverage (privately insured age 44–64 years and Medicare patients age 65 and older). Since the variables are not identical for the groups, each population was examined separately. We examined all records for 2010 through 2020 for prescriptions filled (6 billion records), and for outpatient visits (9 billion records). We extracted all prescription records for A1AT using the generic NDC codes 111973 and 128475, and all outpatient infusions of A1AT (using procedure codes J0256, J0257 and S9346). All unique patients who received at least 30 days of A1AT therapy were identified and their outpatient data from a minimum of 180 days prior to the use of A1AT (Baseline period) extracted. Comparison cohorts of patients were constructed from patients without any use of A1AT who had a diagnosis of emphysema. We used ICD-9 diagnosis codes 492.8 and 492.0 or ICD-10 code J43.9 as appropriate for the data year and extracted all prescription records for A1AT (using the generic NDC codes 111973 and 128475) as well as outpatient infusions of A1AT (using procedure codes J0256, J0257 and S9346). We assigned a 180 day “baseline observation period” to patients and excluded any patients with an AMD diagnosis during baseline (using ICD-9 and ICD10 for codes unspecified macular degeneration (362.5 and H35.30), dry AMD (362.51 and 35.31xx), and wet AMD (362.52 and 35.32xx). For each insurance cohort, we matched the A1AT-exposed patients 1:5 to the emphysema control patients using baseline variables age, sex and comorbidity rheumatoid arthritis diagnosis during the baseline period. We used SAS Proc PSMATCH with a greedy algorithm and a caliper match of 0.2. The two study cohorts of A1AT-exposed patients and their controls were well matched as their standardized differences (green circles, Appendix A) fell within the prespecified cutoff region.

### 4.9. Statistical Analysis

Statistics for mouse and cell culture studies were performed using the ANOVA and *t*-tests after testing for data normality with GraphPad prism software. All data analyzed with the *t*-test or ANOVA passed the normality test (with either the Shapiro–Wilk test, D’Agostino & Pearson test, or the Kolmogorov–Smirnov test). Statistical significance of the data which did not pass the normality test were analyzed using the Mann–Whitney non-parametric test. Data are represented as mean ± SEM. *p* < 0.05 is considered significant. The number of samples used, and further statistical details are added in the figure legends for each experiment.

## Figures and Tables

**Figure 1 cells-12-01308-f001:**
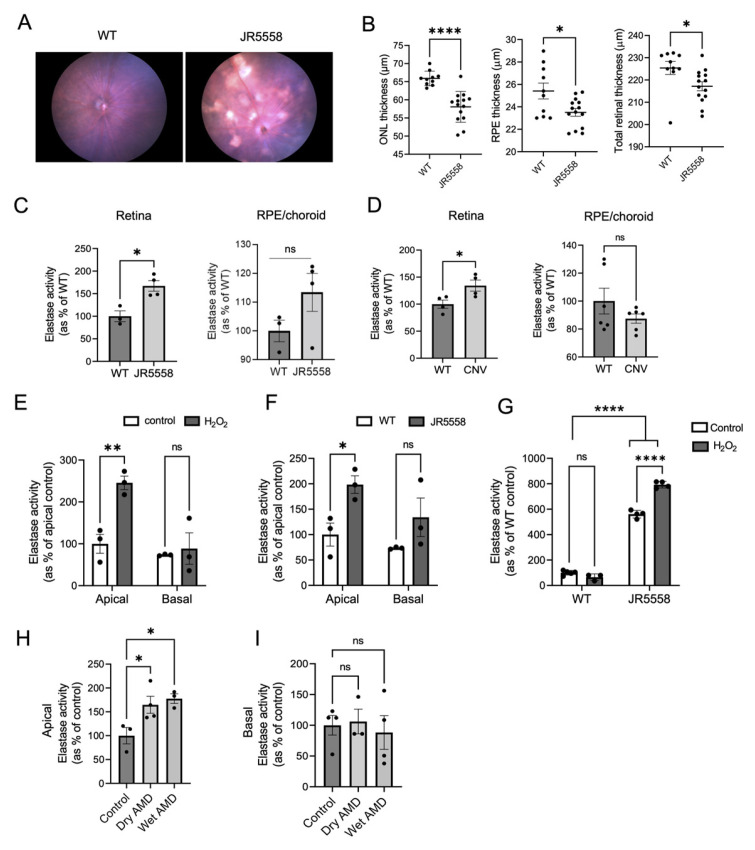
Elastase activity in mouse and cell models of AMD. Fundus images (**A**) and OCT analysis (**B**) of JR5558 and WT mice retinas (*n* = 5–7 mice per group). (**C**) Elastase activity in retina and RPE/choroid extracts of JR5558 and age matched control WT mice (*n* = 3–4 mice per group). (**D**) Elastase activity in retina and RPE/choroid extracts of laser CNV mouse model in comparison to controls (*n* = 4–6 eyes per group). (**E**) Elastase activity in the apical and basal supernatants of control and H_2_O_2_-treated primary WT RPE cells. (**F**) Elastase activity detected in the apical and basal supernatants of primary WT and JR5558 RPE cultures. (**G**) Intracellular elastase activity detected in H_2_O_2_- and non-treated RPE monolayers from WT and JR5558 mice. (**H**) Apical and (**I**) basal elastase activity in human iPSC-derived RPE cells. For all the in vitro assays 3–4 replicates per group were analyzed. Statistics used: (**B**–**D**) *t*-test; (**E**–**G**) two-way ANOVA, Sidak’s MC; * *p* < 0.05, ** *p* < 0.01, **** *p* < 0.0001, ns (not significant). Data represent the mean ± SEM. Abbreviations: WT: wild type (C57BL/6J mice), ONL: Outer Nuclear Layer, RPE: Retinal pigment epithelium.

**Figure 2 cells-12-01308-f002:**
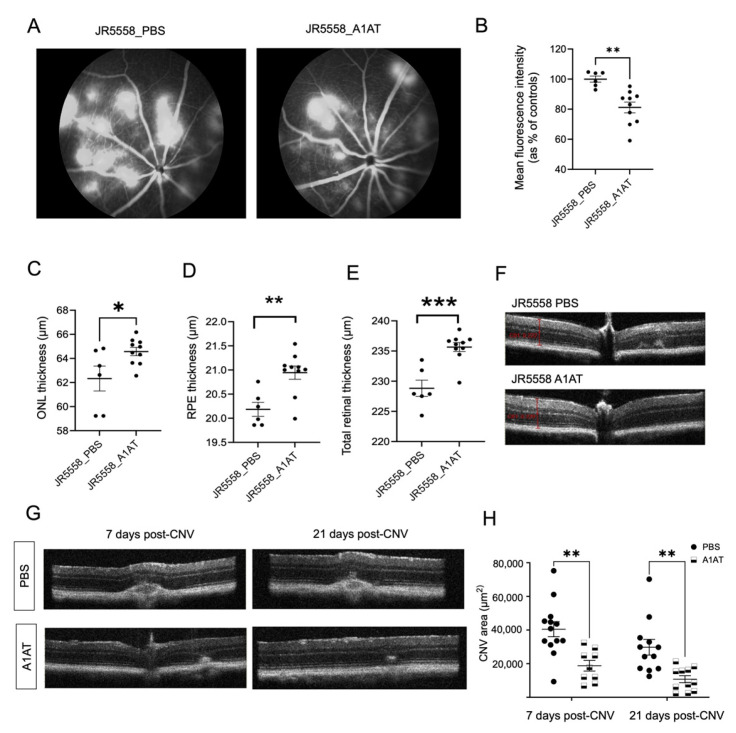
A1AT ameliorates wet AMD pathology in two mouse models. (**A**) Fluorescein angiography (FA) images from PBS- and A1AT-injected JR5558 mice. (**B**) CNV lesion size quantification from FA images. Quantification of ONL (**C**), RPE (**D**) and total retinal thickness (**E**) from the OCT images of PBS and A1AT injected JR5558 mice (*n* = 6–10 eyes per group). (**F**) Representative OCT B-scans of PBS injected and A1AT injected JR5558 mice. Representative OCT images (**G**) showing CNV lesions (post 7 days and 21 days after inducing laser photocoagulation) in PBS- and A1AT-injected laser mice, and the quantification of CNV areas (**H**) from the OCT images (*n* = 9–13 eyes per group). Statistics used: *t*-test for single comparisons, two-way ANOVA, Tukey’s MC for grouped comparisons. * *p* < 0.05, ** *p* < 0.01, *** *p* < 0.001. Data represent the mean ± SEM.

**Figure 3 cells-12-01308-f003:**
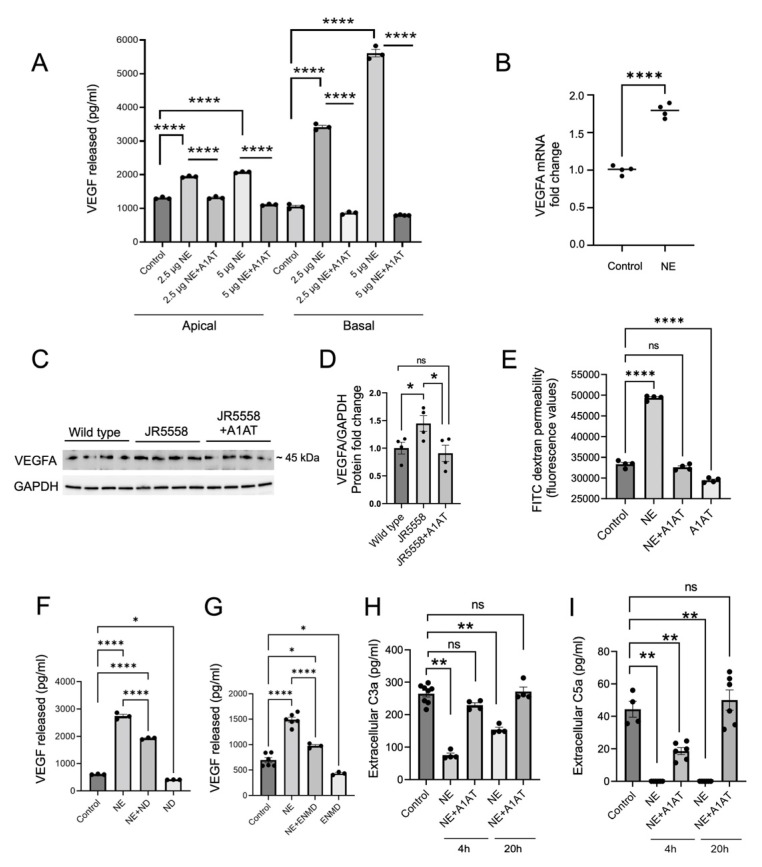
A1AT prevents increase in VEGF levels induced by elastase. (**A**) ELISA-based detection of extracellular VEGF protein levels in the apical and the basal supernatants collected from ARPE-19 monolayers treated with neutrophil elastase (NE) and A1AT for 24 h (*n* = 3–4). (**B**) VEGFA mRNA expression in ARPE-19 monolayers treated with NE for 3 h (*n* = 4) (**C**) Western blot detection of VEGFA protein and its quantification (**D**) in the retinas of WT, JR5558 and A1AT injected JR5558 mice (*n* = 4 retinas per group). (**E**) FITC dextran permeability assay in NE- and A1AT-treated ARPE-19 monolayers (*n* = 4) (**F**) ELISA-based VEGF protein level analysis in the basal supernatants collected from NE and VEGF receptor inhibitor ND (Nintedanib) treated ARPE-19 cells (*n* = 3–4). (**G**) ELISA-based VEGF protein level analysis in the basal supernatants collected from NE and PAR 2 antagonist (ENMD 1068) treated ARPE-19 cells. Extracellular C3a (**H**) and C5a (**I**) protein levels in NE and A1AT treated ARPE-19 cells (*n* = 3–4). Statistics used: (**A**–**G**) *t*-test for single comparison, one-way ANOVA, Dunnett’s MC; (**H**,**I**) Mann-Whitney non-parametric test. * *p* < 0.05, ** *p* < 0.01, **** *p* < 0.0001, ns: not significant. Data represent the mean ± SEM.

**Figure 4 cells-12-01308-f004:**
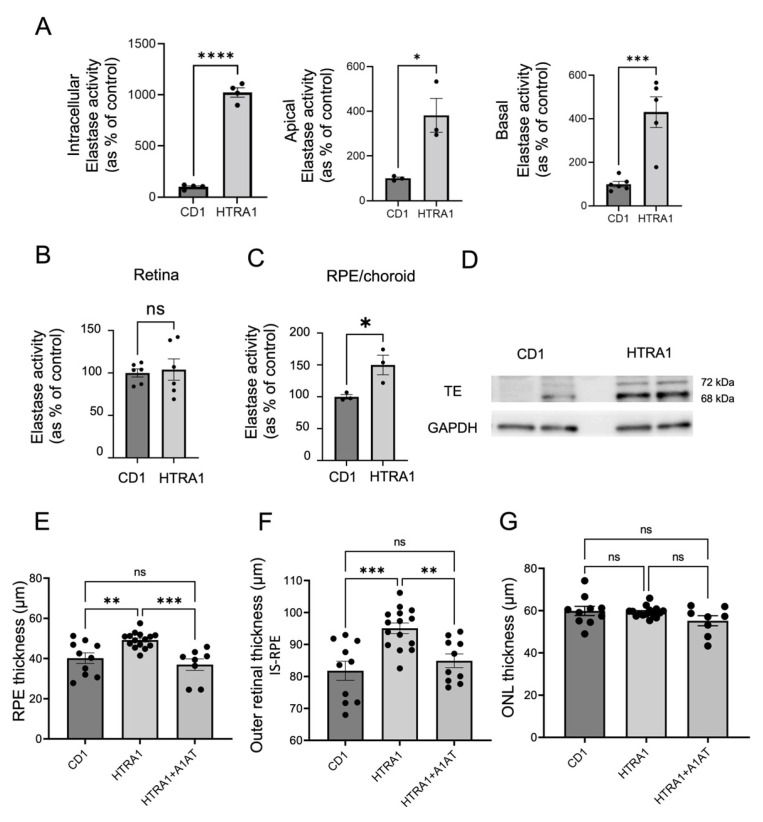
A1AT ameliorates RPE/BrM and outer retinal thickening in HTRA1 transgenic mice. (**A**) Quantification of intracellular, apical, and basal elastase activity in primary RPE cell monolayers derived from HTRA1 when compared to CD1 control mice (*n* = 3). (**B**) Elastase activity in retina and (**C**) RPE/choroid tissue extracts of 6 months old CD1 and HTRA1 mice (*n* = 3). (**D**) Tropoelastin (TE) protein levels present in the RPE/Choroid of 6 months old HTRA1 and CD1 mice. (**E**) OCT quantification of RPE/BrM, (**F**) outer retinal (IS-RPE) and (**G**) outer nuclear (ONL) layer thicknesses in CD1 and HTRA1 mice compared to HTRA1 mice treated with A1AT (*n* = 8–16 eyes per group). Statistics used: *t*-test for single comparison, one-way ANOVA, Tukey’s MC. * *p* < 0.05, ** *p* < 0.01, *** *p* < 0.001, **** *p* < 0.0001, ns: not significant. Data represent the mean ± SEM. Abbreviations: TE: tropoelastin, IS-RPE: inner segments to RPE distance measured.

**Figure 5 cells-12-01308-f005:**
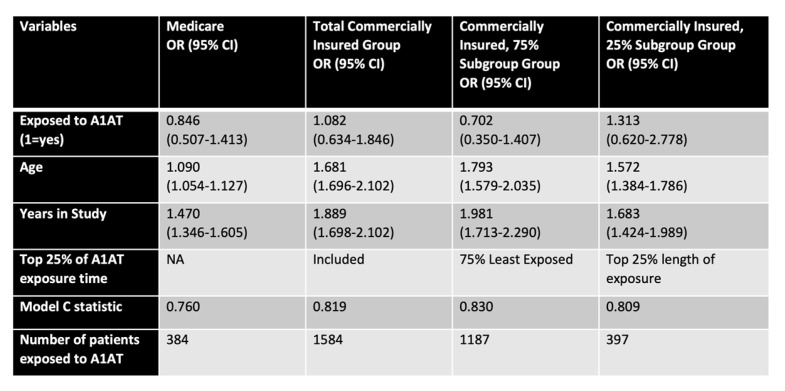
A1AT effects in human subjects. Risk of AMD for matched cohorts of emphysema subjects treated with A1AT or untreated, comparing subjects on Medicare (>65 years of age) or insured commercially (45–65 years of age). The commercially insured cohort is present as all, or divided into two subgroups based on exposure time (lower 75% and top 25%). Variables of analysis included exposure to A1AT, age, years in the study, and Model C statistics.

## Data Availability

Data will be made available upon request.

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
