# Peer review of "Exploring the Therapeutic Potential of Elastase Inhibition in Age-Related Macular Degeneration in Mouse and Human"

_cells, 2023, doi:10.3390/cells12091308_

Round 1
Reviewer 1 Report
Navneet and colleagues report on novel data regarding the potential pro-angiogenic role of increased elastase activity in the progression of AMD and propose alpha-1-antitrypsin (A1AT) as a therapeutic agent for AMD, specifically the neovascular form (wet AMD). In brief, the authors measured elastase activity in retinal lysates from two mouse models of wet AMD, analysed elastase activity in the apical culture medium of primary RPE cells from the same animals and, in addition, included experiments with iPSC-RPE from AMD patients. As a logical consequence of their findings, the authors further focused on A1AT, a serine protease inhibitor, which inactivates elastase and determined its therapeutic effects. The authors also correlated the activity of commercial neutrophil-derived elastase with elevated basal VEGF levels and increased monolayer permeability, suggesting a possible reversible effect of A1AT. They also tested inhibitors of the VEGF receptor and protease-activated receptor 2, emphasizing that elastase-dependent VEGF release is partly controlled by these receptors. Navneet and colleagues then measured elastase activity and the effect of A1AT in retinal tissue lysates and primary RPE cultures from HTRA1-overexpressing mice and discussed a therapeutic effect of A1AT protease inhibitor on HTRA1-mediated retinal thickness and endothelial sprouting. The article concludes with an analysis of AMD risk in pulmonary emphysema patients treated with A1AT, where the authors report a reduced risk of AMD associated with the A1AT treatment.
The manuscript by Navneet and colleagues is of considerable interest and covers an interesting area of AMD pathogenesis and potential treatment options. The article is well written, although not fully structured in the results section which makes it hard to read due to the large number of experiments with numerous methods and models. There are additional weak points in the manuscript, which the authors are invited to revise:
(1) Most of the experiments performed in this study are based on the use of the EnzChek Elastase assay kit, which measures elastase activity by digestion of purified recombinant elastin protein. The results of this study would ideally be supported by a more direct examination of the effect of increased elastase activity on the deposition and degradation of endogenous elastin in the extracellular matrix (ECM), the site where alterations in this structural component have been implicated in the pathogenesis of AMD. Immunofluorescence analysis of elastin fibers in the ECM of primary mouse or human (iPSC-derived) RPE cells may provide a more accurate indication of the effect of such increased elastase activity on elastin turnover. This could also provide an additional platform to test the efficacy of elastase inhibitors. Western blot analysis of soluble tropoelastin in retinal and RPE/choroid cell lysates, as performed in this study, appears not sufficient to reflect a native organisation of elastin fibers in the RPE cellular microenvironment.
(2) The 'Materials and Methods' section needs careful revision as it is sparse with information at some points where a more detailed description would be helpful to the reader. The authors are encouraged to provide information on the protocol, the methods used to measure and analyse the results of the permeability assay and the sprouting assay, which is currently missing.
Further details on cell cultivation time, chemical incubation time, apical or basal chemical application would also be relevant. In section 4.5, there could also be a clear description of the elements considered for the identification of AMD patients (e.g., the specificity of the billing codes used, the source of the data and others).
(3) Authors are asked to provide evidence of prior testing of data normality to justify the use of parametric statistical tests such as ANOVA and t-tests. In situations where data are not normally distributed, non-parametric tests are recommended.
(4) Authors are advised to present their results without bias, interpretation, or judgement. Therefore, the use of terms such as "exciting" (line 166) as well as explicit assumptions (e.g., in lines 91-92 or 127-128) should be omitted and may be more appropriate in the Discussion section.
(5) To facilitate reading, it would be helpful if the numbering of the figures followed the same sequential logic with which the results are presented in the text (e.g., integration of Figure 5 A and B into Figure 1 may be advised).
(6) Supplementary figures and tables need more information, at least a heading and a legend, to enable the reader to understand their content without having to refer to the text of the manuscript.
(7) In line with the above, the authors are asked in Supplementary Figure 1 to indicate the molecular weights (KDa) relative to the Western blot fractions shown and to indicate with an arrow the molecular weight species quantified. In addition, it would be appropriate to show a blot that is as representative as possible of the relative quantification; in particular, the blot in Supplementary Figure 1 for retinal lysates shows almost similar levels of tropoelastin between WT and JR5558 mice, which intuitively does not reflect the relative quantification shown. The number of replicates for this analysis should also be given.
(8) If possible, the authors should provide TER values for iPSC-RPE cells involved in this study. The use of this cell line should be accompanied by characterisation of correct polarisation and expression of RPE differentiation markers in the supplementary material (e.g., by immunocytochemistry).
(9) The authors analysed the apical and basal levels of secreted VEGF in ARPE-19. However, it is well known that these cells, unlike iPSC-RPE cells, do not polarise and achieve high levels of epithelial resistance. The authors should comment on their choice of the cellular model.
(10) Referring to Figure 4, in section 2.7 (line 258), the authors describe a reduction in RPE and outer retinal thickness by A1AT in HTRA1 transgenic mice. The authors should demonstrate (and comment) statistical significance of these effects by comparing treated and untreated HTRA1 transgenic mice.
(11) For consistency of terminology, please refer to Supplementary Fig. 1 as Fig. S1 (lines 137 - 138).
(12) Line 125: It may be appropriate to refer to AMD pathology in general and not specifically to wet AMD, as an increase in elastase activity is reported to be significantly elevated in iPSC-RPE from patients also with dry AMD in the same paragraph.
(13) The reference to Figure 2H is missing both in the text (line 171) and in the legend of Figure 2.
(14) Line 237: Please replace "Fig. 4E" with "Fig. 3E"
(15) Line 186: "…secreted VEGF protein, secreted from RPE" may be replaced by "secreted VEGF protein from RPE".
(16) Line 227: "VEGF" spelt twice, please remove one
(17) Lines 130-131: Please add appropriate references in the following sentence “In AMD patients, elastin fragments and autoantibodies are increased; and antibody-mediated complement activation has been proposed to play a role in pathology”.
(18) End punctuation (period) could be removed from the main title and paragraph headings
Reviewer 2 Report
The manuscript “Exploring the therapeutic potential of elastase inhibition in age-related macular degeneration in mouse and human” is an interesting and potentially important series of observations using an elastase inhibitor to impact AMD pathobiology. The manuscript is long, detailed, with many figures and supplemental information to make the argument that A1AT, an elastase inhibitor, may be of use in treating wet AMD. The insurance database results are potentially the most interesting but need to include the fact that A1AT is used to treat emphysema, a disease frequently associated with smoking - a known risk factor for AMD. The discussion of the different risk groups based on age needs to include the potential that maybe they were younger because they smoke more, for longer, and that might impact AMD onset. Finally, AMD is racially biased, and the potential for impact on insured and uninsured/groups should be included and discussed.
Overall, the broad use of diverse models systems, including human data, strengthens the presentation despite some weaknesses, described below.
Specific comments:
Induced HPSC RPE are fetal, explain and justify the use fetal cells as an AMD model. These cells have never digested a photoreceptor outer segment, a primary function of RPE. They have not endured 71 years of daily challenge. Two lines of fetal cells, coupled with ARPE-19 severely limits interpretation. The inclusion of transgenic animals really strengthens the whole manuscript, and helps get past the RPE cell culture issue.
Please include a phase-contrast image of the cells, were they polygonal, polarized, and pigmented? Were they postproliferative?
Section 2.2. Revise to clarify RPE cells from transgenic animals vs. iPSC RPE. Clarify how monolayers were judged as ‘stable’. Barrier function listed at 200 ohm*cm2 seems low, and the barrier listed for ARPE19 suggests to me why they are not a good model for RPE studies. The authors' conclusion that they were confluent, with a barrier under 50 ohms*cm2 seems questionable for confluent RPE.
ARPE-19 are a significant weakness for AMD studies. The cells are typically not post-proliferative, polarized, pigmented, and polygonal. Please justify the use of ARPE-19, and the stated minimal barrier properties.
A1AT inhibited monolayer integrity loss please show cells. This is not really demonstrated. Integrity of the RPE monolayer was assessed by TER, and wells with TER of ~40-45 Ω*cm2were used for the experiments. This is really low for RPE. Suggesting the cells were just confluent, and were not differentiated, pigmented, and really functional as an RPE monolayer.
iPSC cells were derived using a Barti protocol, but ‘aging’ is not discussed. As above, using fetal cells to model an aging disease is a significant weakness.
For figure 5 C. Add actual patient ages.
Line 405: PMID: 22408008. delete and revise.
Reviewer 3 Report
In this manuscript authors reported an increase in elastase activity in the retinas and RPE cells of AMD mouse models, and AMD patient-iPSC-derived RPE cells. Moreover, they found that A1AT reduced CNV lesion sizes in mouse models while A1AT inhibited elastase-induced VEGF expression and secretion, and restored RPE monolayer integrity in ARPE-19 cells. Finally, authors found that A1AT mitigated RPE-BrM thickening in HTRA1 overexpressing mice.
The study is very interesting, generally well written and well illustrated. However, some points deserve to be improved. In particular:
Introduction: HTRA1 introduction is very poor and its multifaceted role deserves to be highlighted since it plays an important function in ECM remodelling (PMID: 28076935), a key process in AMD pathogenesis (PMID: 36204224).
Lines 47-53: References are needed
Figure 3: "picograms" is not an appopriate measure unit for ELISA. Do the authors mean pg/mL?
Figures: When Western blots are shown, molecular weights must be reported
Line 405: PMID must be replaced with reference
Abbreviations must be written in full length when mentioned for the first time
Round 2
Reviewer 1 Report
The authors have made substantial revisions to the original manuscript and, by doing so, have well responded to the suggestions and comments of the reviewer. Overall, the manuscript has gained in clarity and readability. A most interesting and remarkable piece of scientific work.